# Understanding the Impact of Chronic Rhinosinusitis with Nasal Polyposis on Smell and Taste: An International Patient Experience Survey

**DOI:** 10.3390/jcm12165367

**Published:** 2023-08-18

**Authors:** Louis Luke, Liam Lee, Shyam Ajay Gokani, Duncan Boak, Jim Boardman, Carl Philpott

**Affiliations:** 1Ear, Nose and Throat (ENT) Department, James Paget University Hospital, James Paget University Hospitals NHS Foundation Trust, Great Yarmouth NR31 6LA, UK; c.philpott@uea.ac.uk; 2Norwich Medical School, University of East Anglia, Norwich NR4 7TJ, UK; 3Fifth Sense, Unit 2, Franklins House, Wesley Lane, Bicester OX26 6JU, UK

**Keywords:** polyps, chronic rhinosinusitis, olfaction, taste, patient experience, hyposmia, anosmia

## Abstract

The aim is to understand the patient experience of living with chronic rhinosinusitis with nasal polyposis (CRSwNP), clinician interactions and how symptoms, smell and taste disturbance are managed. An anonymized, online survey was distributed through a UK charity, Fifth Sense, a UK otolaryngology clinic and online support groups to capture qualitative and quantitative data. Data were collected from 1st December 2022 to 1st February 2023. A total of 124 individuals participated. The majority were female (66%) and in the age range of 41–70 years; 74.2% of participants were from the UK with the rest from North America, Europe and Asia. A total of 107 participants declared they had CRSwNP. Rhinologists and general otolaryngology clinicians scored the highest for patient satisfaction whilst general practitioners scored the lowest. Satisfaction with the management of smell and taste disturbance was lower amongst all clinicians compared to overall satisfaction. Ratings correlated with response to therapy and clinician interactions. Respondents reported hyposmia/anosmia to be the most debilitating symptom. Surgery and oral steroids were considered to be effective; however, the benefit lasted less than six months (62%). Hyposmia/anosmia is a key CRSwNP symptom that has limited treatment options and is frequently undervalued by clinicians. There is a need for more effective management options, education and patient support.

## 1. Introduction

Chronic rhinosinusitis (CRS) is a common condition with a prevalence of 10% in the UK population [1]. CRS has a significant detrimental impact on quality of life, large associated healthcare resource use and individual costs [2,3]. CRS in adults is defined as the presence of two or more symptoms for 12 weeks or longer that must include nasal obstruction/blockage/congestion or rhinorrhea (anterior/posterior) with or without facial pain/pressure and/or hyposmia/anosmia. Traditionally, CRS has been classified into CRS with nasal polyposis (CRSwNP) or CRS without nasal polyposis (CRSsNP) [4]. However, the update from the European Position Paper on Rhinosinusitis and Nasal Polyps (EPOS) 2020 has moved away from this terminology and focused on CRS as a spectrum of different conditions. Patients are divided into primary and secondary CRS and are then further divided into localized and diffuse disease. In primary CRS, there is a focus on determining endotype dominance (type 2 or non-type 2) and, from that, the phenotype [1].

To diagnose CRSwNP, there must be evidence of diffuse, bilateral nasal polyps seen either on nasal endoscopy or on a computed tomography (CT) scan [1]. Roughly 25–30% of CRS patients have CRSwNP [5]. Banerji et al. found that sinonasal symptoms such as nasal obstruction and reduced or loss of smell were significantly associated with CRSwNP compared to CRSsNP [6]. Furthermore, patients with CRSwNP reported higher nasal domain symptom scores on the SNOT-22 questionnaire, preoperative CT scores and endoscopic scores compared to CRSsNP [6,7,8].

The mainstay of management for diffuse, bilateral CRS regardless of polyps or without polyps are saline sinus rinses and local corticosteroids. EPOS 2020 emphasizes the importance of having an integrated care pathway in the management of CRS, which includes the avoidance of exacerbating factors and the avoidance of antibiotics. Other medical therapies that can be used include a short course of a systemic corticosteroid, short course of broad-spectrum/culture-directed antibiotics or prolonged courses of low-dose anti-inflammatory antibiotics. Endoscopic sinus surgery (ESS) is the next step after appropriate medical therapy. For patients with type 2 endotypes that are resistant or not fit for ESS and fit certain criteria, the use of monoclonal antibodies can be used to manage CRS but are not yet approved by the National Institute of Clinical Excellence (NICE) for CRS in the UK [1].

Olfactory disturbance has a prevalence as high as 60–80% in CRS patients [9]. It is well recognized that olfactory and gustatory disturbance can have a profound negative impact on a patient’s quality of life including mood, relationships, personal safety, nutrition and physical health [10,11,12]. Nasal polyposis, age, smoking, asthma and eosinophilia were associated with olfactory dysfunction in CRS patients [13,14]. Currently, there is a paucity of research on the experiences of CRSwNP patients in relation to smell and “taste” disturbances.

The aim of this study is to explore the UK and international experiences of patients in living with and managing CRswNP with healthcare professionals in addition to understanding the impact on smell and taste. In conjunction with the charity, Fifth Sense, we designed and distributed an online survey to explore these themes.

## 2. Materials and Methods

### 2.1. Study Design

This study is a cross-sectional, online survey exploring the experiences of smell and taste in patients with CRSwNP. This survey was created using Microsoft Forms to gather quantitative and qualitative data from CRSwNP participants. The survey included multiple choice questions, rating scales and free text answers and was therefore both quantitative and qualitative in nature. Questions were designed with feedback and support from an olfactologist and two patient representatives from the Fifth Sense charity organization. The concept of the survey was started on 7th November 2022. The final survey design was completed on 30th November 2022 after being reviewed by all authors. The survey was distributed through Fifth Sense email newsletters, social media channels and patients presenting to a UK rhinology clinic from 1st December 2022, and survey responses were collated up to and including 1st February 2023.

### 2.2. Setting

The survey was distributed through Fifth Sense email newsletters, social media channels and patients presenting to a UK rhinology clinic. It was open to anyone who could access it online. The survey was free to access and voluntary to participate in.

### 2.3. Participants

Any individual with a self-reported medical diagnosis of chronic rhinosinusitis with nasal polyposis (CRSwNP) was eligible to participate in the survey and was included. Participants who reported that they did not have a diagnosis of CRSwNP were excluded from the study. There were no other exclusion criteria. A total of 124 participants completed the survey, and 107 reported a diagnosis of CRSwNP, which was analyzed.

### 2.4. Data Sources and Variables

Data collected from the survey included general demographics (gender, age group, location of residence and whether they had a formal diagnosis of CRSwNP), experiences of living with CRSwNP and the effect on their smell and taste. The full survey is available at https://forms.office.com/r/6Q0Lpr5ZWm (accessed on 12 July 2023).

### 2.5. Bias

To reduce selection bias in this study, the survey was distributed online through a UK-based charity organization, a UK otorhinolaryngology clinic and social media channels including patient support groups to capture respondents from all countries and backgrounds.

### 2.6. Study Size and Statistical Analyses

There was no minimum sample size required for this study as only descriptive statistics were performed. All figures and tables were created using R using the ggplot2 package [15].

### 2.7. Ethical Approval

As the survey was anonymous and considered to be a service evaluation, there was no ethical approval sought in line with the Health Regulation Authority guidance:

https://www.hra-decisiontools.org.uk/research/docs/definingresearchtable_oct2017-1.pdf (accessed on 12 July 2023). No patient-identifiable information was collected as part of the study. The survey contained a statement, “We hope to publish the results from this survey as an original research article and present the results at scientific meetings/conferences”, to make clear the intentions of the use of anonymized data gathered.

## 3. Results

### 3.1. Demographics

Patient demographics are summarized in Table 1. Of the 107 participants, the majority were female (65.4%), and the rest were male (34.6%). The most common age groups were between 41 and 55 and 56 and 70, accounting for 67.2% of respondents. The majority of respondents were based in the UK, and 22.4% were from outside the UK, including participants from Australia, Canada, Hungary, Iceland, the Philippines, South Africa, Sweden and the USA. This group was asked further questions in the survey whilst the survey finished for the 17 non-CRSwNP respondents.

### 3.2. Patient Experience of Living with CRSwNP

Patient experiences were gathered for those who declared they have been formally diagnosed with CRSwNP as shown in Table 2. The majority of participants (81.3%) had suffered with symptoms of CRSwNP for over five years with 69.2% being formally diagnosed for over five years. A total of 97.2% were diagnosed by general otorhinolaryngology (43.1%) or otolaryngology clinicians specializing in rhinology (36.9%); 66 participants (61.7%) declared they were not under regular follow-up. For the 56 participants who were under regular follow-up, the majority (78.5%) were under the care of an otorhinolaryngology clinician. There was a range of responses on what participants felt could be carried out to improve their experience or management of their condition. The majority felt more management options were needed (74.7%) followed by smell testing (39.4%), more time with the clinician (29.3%) and mental health support (24.2%).

Other reasons included having better health insurance coverage for treatments in the USA:

‘Costs for treatments are massive here with sublingual immunotherapy and Budesonide, Flovent, and the potential for Dupixent are all expensive and poorly covered by my insurance.’(Male, 26–40, USA).

Conversely, UK respondents would have liked more management options such as Dupixent (Dupilumab, a monoclonal antibody) that are available in the USA and not licensed for use in the UK National Health Service (NHS):

‘I find it very frustrating that we can’t get the biologics for such a severe illness. I know my ENT doctor is trying his best, just limited by the medications he can use.’(Female, 56–70, East Midlands, UK).

In addition, further education amongst non-otolaryngology clinicians such as general practitioners (GPs) to manage the condition was mentioned:

‘Primary care physicians and allergists require education to manage initial symptoms and referral to competent specialists.’(Female, 56–70, USA).

‘More awareness from the GP of CRSwNP.’(Female, 56–70, South East England).

Figure 1 demonstrates the average satisfaction ratings of CRSwNP management and management of smell and taste they experienced from different healthcare professionals. The highest satisfaction scores were seen by rhinologists followed by a general ENT doctor, respiratory physician/allergist/immunologist, GP and other. Participants had the opportunity to provide an explanation in a free text box for their scores (Table 3).

Some of the reasons highlighted for the high scores amongst rhinologists and general ENT doctors were effective treatment and having a knowledgeable and attentive doctor.

However, there were some respondents who were dissatisfied with the overall management by their general ENT clinician or rhinologist. Reasons given included the recurrence of symptoms, limited treatment options and long time periods between appointments, investigations and treatments such as surgery.

In contrast to satisfaction with CRSwNP management, respondents had differing experiences when seeing rhinologists. Some participants felt that dealing with reduced/loss of smell was not the clinician’s priority, and others felt their rhinologist focused on their sense of smell. Those who were seen by respiratory/allergy/immunologist teams provided mixed scores depending on their interaction with their clinician.

Positive responses in the GP group were often associated with a referral to a specialist or whether they focused on symptoms including the loss/reduced sense of smell symptom.

Negative responses had a recurrent theme of lack of knowledge, repetitive treatments that did not work and a lack of empathy/interest in their condition. Ratings were lower in the management of smell and taste impairment. Responses from the ‘other’ group provided the lowest rating scores, and these were all related to not having any follow-up for their condition.

Overall satisfaction with clinicians managing smell and taste impairment was lower compared to satisfaction with CRSwNP management (Figure 1). These results may be explained by hyposmia/anosmia being difficult symptoms to improve with current therapies and/or clinicians underappreciating the impact of olfaction on quality of life. Satisfaction correlated with response to treatment and patient perceptions of how their condition is managed and the personal interactions with their healthcare professionals.

Figure 2 demonstrates how participants would rate the personal impact of each of the key symptoms of chronic rhinosinusitis [1]. Overwhelmingly, 76 of the 107 respondents (71%) felt that a loss/reduced sense of smell was the most debilitating symptom, stating their problem to be as bad as it can be, and 0 stated that there were no problems. Nasal blockage/obstruction was the next troublesome symptom amongst participants as 22 (20.6%) gave a rating of 5 and 42 (39.3%) a rating of 4. Rhinorrhea (post-nasal drip and runny nose) was a moderate issue with facial pain/pressure having varying levels of impact.

Respondents were asked which treatment they were currently using and what treatment they perceived to be the most effective in providing symptomatic relief (Figure 3). The most popular treatments were nasal douching and nasal steroid sprays. However, only three and nine people found them effective in treating symptoms, respectively. In contrast, oral steroids and sinus surgery were the top two effective management options. Other effective treatments included aspirin desensitization, specific brand names for nasal steroid sprays and injectable monoclonal antibody therapies. Despite certain therapies being more effective than others as described, 62% only had symptomatic relief for less than 6 months; 34 people (36.4%) reported side effects related to their treatment. Common answers were nosebleeds due to dryness/crusting from either nasal sprays or surgery (6.5%), headaches after using nasal steroid sprays (1.9%) and a range of side effects from the use of oral steroids such as increased appetite, lack of sleep or lethargy (2.8%). Figure 3 compares regular therapies and their perceived limited efficacy in symptomatic relief. Oral steroids and sinus surgery are effective management options that cannot be used on a regular basis due to side effects and potential serious risks and complications. This may be explained by respondents having severe CRSwNP that is non-amenable to regular medical therapy.

### 3.3. The Effect of Living with CRSwNP on Smell and/or Taste

Participants were asked if they had any concomitant olfactory disorders in addition to CRSwNP; more than one option could be selected (Table 4). The majority (60.6%) of the total respondents only had CRSwNP, with five respondents providing no answers. A total of 38 participants (35%) declared they had other conditions that affected their sense of smell. Of these, 73.7% had other sinonasal conditions that included allergic rhinitis, nasal tumors or olfactory cleft stenosis, 15.8% after an operation and 7.9% after COVID-19/viral infection.

Figure 4 shows how participants rate their sense of smell and flavor perception (blend of taste and smell sensations evoked by a substance in the mouth) before being diagnosed and after receiving treatment for CRSwNP. This is to demonstrate the change in perceptions before starting treatment once a diagnosis of CRSwNP has been established and perceptions after regular treatment. A total of 44% of participants rated their smell and flavor perception to be ‘good’ or ‘very good’ before diagnosis, but this declined to 20% and 23%, respectively. There was an increase in ‘poor’ or ‘very poor’ responses in smell and flavor perception after treatment, which shows persistent or worsening smell and flavor perception despite therapies. This may be due to the limited or poor efficacy of therapies in managing olfactory disturbance or due to the natural disease progression of CRSwNP.

Figure 5 highlights what the impact of reduced/loss of smell has on different aspects of life for each participant. These results are in keeping with the literature on the impact of olfactory disturbance on quality of life [10,11,12]. Besides the impact on occupation and relationships, the majority answered that loss/reduced sense of smell ‘often’ or ‘always’ impacts the following areas of life in descending order: enjoyment of food/drink, food safety, gas/smoke safety, personal hygiene, mood, hygiene of children/pets and sleep.

## 4. Discussion

### 4.1. Key Findings

This study of 107 participants living with CRSwNP provides insight into the management of their condition and use impact on their smell and taste. The highest satisfaction with management was among rhinologists and general ENT clinicians, while the lowest was with GPs. Reasons for high satisfaction included seeing a knowledgeable clinician with effective treatment and having an empathetic and honest clinician. On the other hand, reasons for low satisfaction were persistent symptoms, limited management options, and long waiting times.

Common management options included nasal douching and intranasal corticosteroids, while sinus surgery and monoclonal antibodies were used the least. Sinus surgery and oral steroids were considered the two most effective therapies, but 62% of respondents reported relief lasting less than 6 months. Side effects included epistaxis or crusting from nasal steroid spray or previous sinus surgery and increased satiety and lack of sleep from oral steroid usage.

Loss or reduced sense of smell was identified as the main, difficult-to-manage symptom of CRSwNP, and it appeared to deteriorate after diagnosis and treatment. The impact of olfactory dysfunction on quality of life was significant, affecting enjoyment of food/drink, safety, personal hygiene, and mood. Participants had lower satisfaction ratings with healthcare professionals managing smell/taste impairment compared to the overall management of CRSwNP. Negative perceptions were due to a lack of interest or disregard for olfactory disturbance by clinicians. On the other hand, positive encounters were due to improvement in smell with treatment or experiences with clinicians who focused on olfaction.

### 4.2. Strengths and Limitations

The strengths of this study include an anonymous and voluntary survey collecting both quantitative and qualitative data on the experiences of smell and taste in patients with CRSwNP. The survey was distributed through a UK-based charity, a UK otorhinolaryngology clinic and social media channels, which allowed for diverse recruitment of participants from different countries and backgrounds. The design was convenient, cost-effective and accessible, reducing barriers to participation and increasing the response rate (total of 124 participants).

However, there are limitations to this study that must be considered when interpreting the results. Firstly, the study relied on self-reported claims about a diagnosis of CRSwNP, which was not verified as this was an anonymous study. Moreover, the majority of respondents were female despite there being evidence that more male patients had CRSwNP which could affect results [5]. There were no strict inclusion or exclusion criteria for respondents to participate in the survey, and co-morbidities, allergies or smoking status were not collected. Conditions that affected smell and taste were collected and grouped together. For example, sinonasal conditions included allergies, olfactory cleft stenosis and sinonasal tumors. However, a subgroup analysis was not performed for these participants as confounding factors were not controlled. A comparison between the responses from the CRSwNP and CRSwNP with additional medical conditions that affect smell groups could have identified differences in perceptions on CRSwNP and olfaction. It is important to appreciate that the study is prone to self-selection bias, as the survey was voluntarily completed by participants. Respondents with strong opinions or experiences with CRSwNP and those particularly affected by smell or taste disturbance would be more likely to participate. In addition, respondents who are members of the Fifth Sense charity or part of social media patient support groups may have been more likely to engage with the survey, leading to a respondent bias in the results. Differences in the management of CRSwNP between different countries made certain interpretations challenging. For instance, there is limited use of monoclonal antibodies for CRSwNP in the NHS, whereas in the USA, monoclonal antibodies such as Dupixent are more readily available [16]. This makes it difficult to evaluate patient perceptions of the efficacy of monoclonal antibodies due to low numbers in this survey despite randomized controlled trial evidence proving their effectiveness for difficult-to-treat CRSwNP cases [17]. Finally, technical limitations such as internet access and computer/English proficiency may have affected the response rate and participation.

### 4.3. Comparison with Other Studies

Erskine et al. identified overlapping themes in their qualitative study of 21 adult CRS patients, including dissatisfaction with prolonged time to treatment and limited or short-lived effects of treatment [18]. Additionally, in line with our study, reduced enjoyment of food and drink and safety concerns were identified as important consequences of olfactory disturbance in other surveys of patients with CRS [19,20].

CRSwNP has a more severe nasal symptom profile in terms of nasal obstruction, rhinorrhea and loss/reduced sense of smell when compared to CRSsNP [6,7]. Talat et al. highlighted that CRSwNP patients report greater hyposmia and significantly less symptom control when compared to CRSsNP patients [21]. The patients in our study identified hyposmia/anosmia to be the most debilitating symptom of CRSwNP. In comparison to other studies, Abdalla et al. found nasal blockage (96.5%) and altered sense of smell/taste (90.3%) to be the most prevalent symptoms in CRSwNP patients undergoing sinus surgery [22]. Conversely, an online survey by Hopkins et al. showed different results with CRS patients considering headache and rhinorrhea to be more important symptoms than smell and nasal obstruction [23]; however, this study was self-reported with those with and without polyps being reported together.

In this survey, sinus surgery and adjunctive oral corticosteroids were highlighted as being effective treatment options in managing overall symptoms including olfactory disturbance. Short-term oral corticosteroids are accessible to patients as these can be prescribed by any clinician. They improve symptom severity and quality of life outcome measures compared to placebo; however, this improvement does not extend beyond three months [24,25]. In addition, adverse effects from their use can occur including sleep disturbance reported by some respondents in the short term or osteoporosis, cataracts and increased susceptibility to infections with prolonged use [26]. Referral to secondary care enables access to a wider array of treatment options, including the potential for endoscopic sinus surgery. Whilst surgery may help to achieve better disease control compared to medical therapy alone, there is the risk of iatrogenic injury to the olfactory epithelial surfaces, which may have been the case for six respondents in our cohort [27,28]. Furthermore, there is a risk of disease recurrence over time with some patients requiring revision surgery at a later stage [29,30]. In a prospective cohort study of 3128 UK patients undergoing sinus surgery for CRS, Hopkins et al. identified that 18% of patients did not receive a pre-operative course of steroid treatment, and 11.4% went on to require revision surgery by 36 months [31]; however, this study is now two decades old, and practices may well have changed. There remains a need for the standardization of practice in the management of olfactory disturbance in CRS to ensure equal opportunities for gold-standard care.

Barriers to effective care highlighted in this study included failure to recognize the gravity of olfactory disturbance, excessive treatment costs and varied treatment quality between primary and secondary care. These issues were also highlighted by Ball et al. in a global cross-sectional study of 673 patients with smell and taste disorders [32]. The study also involved self-selecting participants but was not restricted to patients with CRS. Ball et al. reported a mean personal cost of GBP 421 to patients for seeking advice and treatment, suggesting that the cost barrier is a worldwide issue, not only in the USA.

### 4.4. Implications for Future Research and/or Practice

The use of targeted therapies such as monoclonal antibodies for CRS, which have strong evidence for effectiveness, is growing [1]. These are being used in the USA, but in the UK, monoclonal antibodies are limited to patients who meet strict criteria or through clinical trials such as the randomized, double-blind, parallel-group, phase III study, ANCHOR-1 [16,33,34]. In the future, monoclonal antibodies may become more widely available in the UK and improve patient symptoms and satisfaction. It will be important to understand the effectiveness and patient experiences of monoclonal antibodies compared to current therapies through clinical trials and qualitative studies, respectively, in order to identify tailored treatments for specific patient groups.

Despite a loss/reduced sense of smell being a common, debilitating symptom of CRSwNP, olfactory/gustatory dysfunction is often under-recognized by healthcare professionals, including ENT clinicians, leading to patient isolation and poor mental health [32]. The findings from this survey highlight these key issues amongst GPs, respiratory physicians/allergists/immunologists and ENT clinicians. Improving olfactory disturbance education is needed for clinicians. Smell testing, olfactory training, and personal safety support, such as improving awareness of gas safety and rotten food through education and support groups, may help manage hyposmia/anosmia and improve patient satisfaction. Future research is needed to understand what the barriers are to addressing smell and taste disorders and how these can be addressed. Effective strategies can then be used to increase public awareness and improve the education of healthcare professionals around olfactory dysfunction.

Olfactory disturbance and CRS have a profound impact on patient psychosocial wellbeing due to lack of clinician knowledge, delays in referral and repetitive ineffective therapies [11,18,32]. Simple measures such as more clinic time and mental health support through support groups or therapies such as cognitive behavioral therapy (CBT) may help patients come to terms with their condition. Further research is needed on the feasibility, efficacy and value of these options to help alleviate the psychological impact of CRSwNP. A combination of psychological and physical treatment modalities in the future may be required to manage patients holistically, especially for those who gain minimal to no benefit from current management options.

## 5. Conclusions

From this survey, we have explored what the experiences are of patients living with CRSwNP, how their smell and taste are managed and their perceptions of healthcare in and outside the UK. A number of key themes have been raised by patients showing reduced/loss of sense of smell to be a key, major symptom amongst patients, which can often be overlooked and undervalued by all clinicians including generalists and specialists. In the future, it is hoped that more management options such as monoclonal antibodies may become available for patients with difficult-to-manage CRSwNP. Whilst patients wait, it may become necessary to help support patients in a holistic manner when managing smell and flavor disturbances and other symptoms of CRSwNP.

## Figures and Tables

**Figure 1 jcm-12-05367-f001:**
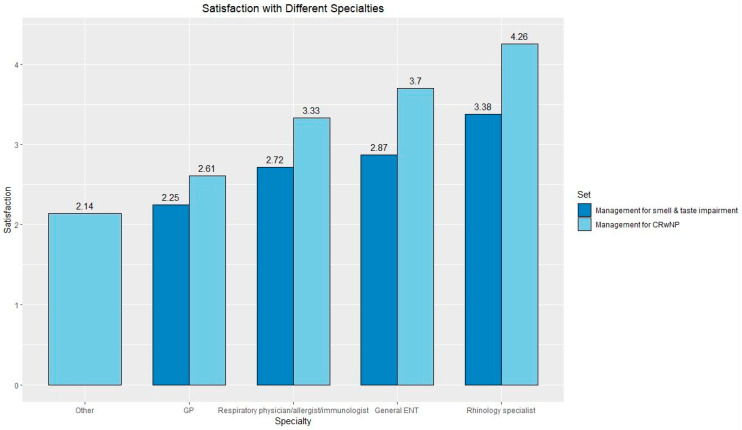
Average satisfaction responses for management of CRSwNP (management of smell and taste impairment). Graded from 1–5 with 5 being highest satisfaction. Scores were as follows—Other 2.14, GP 2.61 (2.25), Respiratory physician/allergist/immunologist 3.33 (2.72), General ENT doctor 3.70 (2.87) and Rhinology specialist 4.26 (3.38).

**Figure 2 jcm-12-05367-f002:**
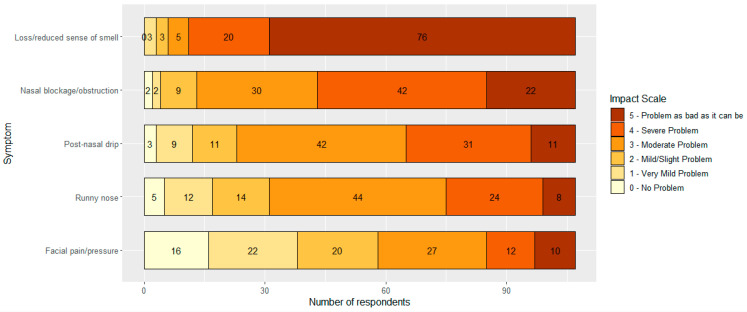
Symptom impact scale. Numbers represent number of respondents who chose the answer.

**Figure 3 jcm-12-05367-f003:**
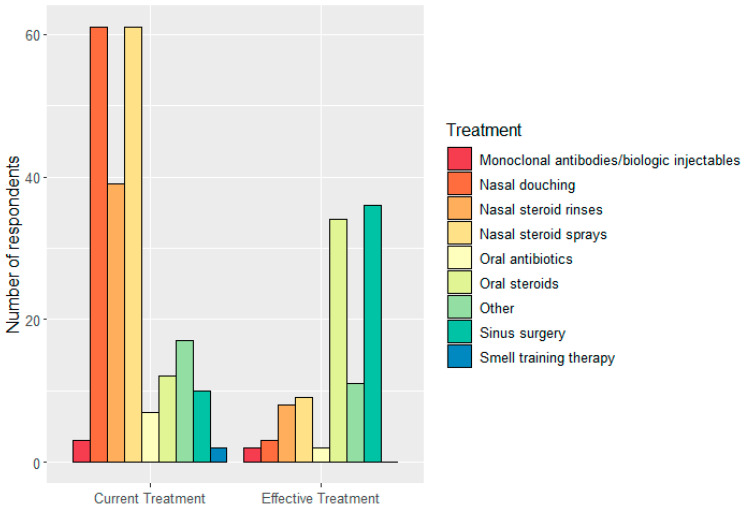
Current treatment and perceived effective treatment. N.B. Patients were allowed to choose more than one answer.

**Figure 4 jcm-12-05367-f004:**
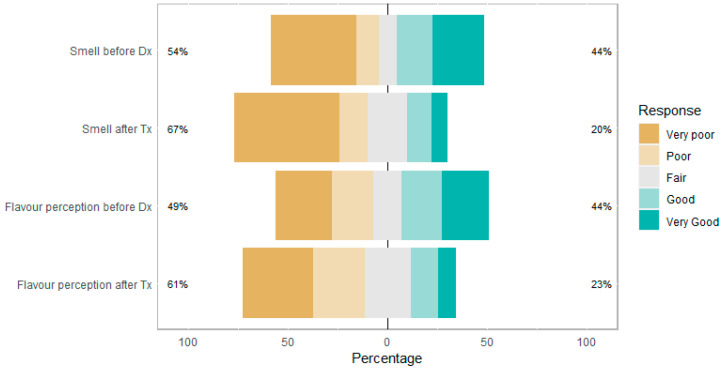
Perception of sense and flavor perception before diagnosis (Dx) and after treatment (Tx). Percentage on each end shows the percentage of responses above and below ‘Fair’.

**Figure 5 jcm-12-05367-f005:**
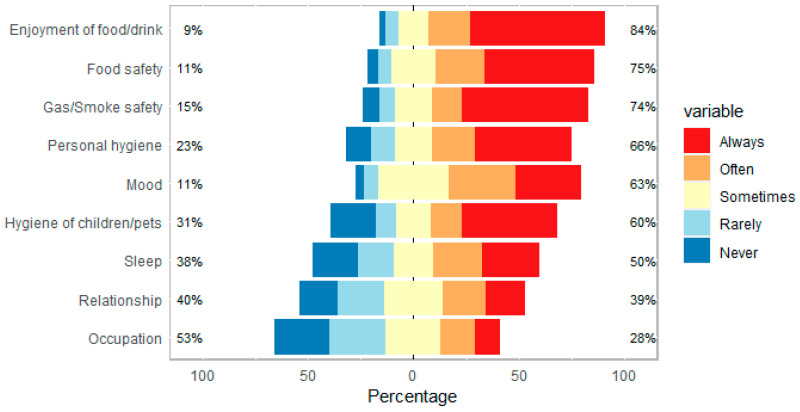
Impact of reduced/loss of smell and taste on various aspects of life. Percentage on each end shows the percentage of responses above and below ‘Sometimes’.

**Table 1 jcm-12-05367-t001:** Participant demographics.

Variable	Frequency (%)
Gender	
Female	70 (65.4%)
Male	37 (34.6%)
Gender neutral	0
Transgender	0
Prefer not to say	0
Age	
Under 18	0
18–25	3 (2.8%)
26–40	20 (18.8%)
41–55	36 (33.6%)
56–70	36 (33.6%)
Over 70	12 (11.2%)
Region	
East of England	8 (7.5%)
East Midlands	4 (3.7%)
West Midlands	6 (5.6%)
Greater London	6 (5.6%)
North West England	9 (8.4%)
North East England	3 (2.8%)
South West England	13 (12.1%)
South East England	17 (15.9%)
Yorkshire and Lincolnshire	6 (5.6%)
Scotland	4 (3.7%)
Wales	7 (6.5%)
Northern Ireland	0
Abroad	24 (22.4%)
Australia (2)	
Canada (4)	
Hungary (1)	
Iceland (1)	
Philippines (1)	
Republic of Ireland (1)	
South Africa (1)	
Sweden (1)	
USA (12)	
Formally diagnosed with CRSwNP	
Yes	107 (86%)
No	17 (14%)

**Table 2 jcm-12-05367-t002:** Patient experiences of living with CRSwNP.

Question	Frequency (%)
How long have you suffered with symptoms of CRSwNP?	
Less than a year	1 (0.9%)
1–2 years	5 (4.7%)
2–5 years	12 (11.2%)
5+ years	87 (81.3%)
Not sure	2 (1.9%)
Who diagnosed you with CRSwNP? (Allowed to choose more than one answer)	
General Practitioner	14 (10.8%)
General ENT doctor	56 (43.1%)
ENT doctor specializing in rhinology	48 (36.9%)
Respiratory physician/allergist/immunologist	9 (6.9%)
Other	3 (2.3%)
How many times have you been referred to a specialist (e.g., ENT) by your GP for CRSwNP?	
1	20 (18.7%)
2–3	40 (37.4%)
4–5	17 (15.9%)
5+	22 (20.6%)
Never been referred	8 (7.48%)
How long since you were initially diagnosed with CRSwNP?	
Less than a year	7 (6.5%)
1–2 years	10 (9.3%)
2–5 years	15 (14%)
5+ years	74 (69.2%)
Not sure	1 (0.9%)
Are you under any regular follow-up for your CRSwNP?	
Yes	39 (36.4%)
No	66 (61.7%)
Not applicable	2 (1.9%)
Who regularly sees you regarding your CRSwNP follow-ups?	
General Practitioner	9 (8.4%)
General ENT doctor	20 (18.7%)
ENT doctor specializing in rhinology	24 (22.4%)
Respiratory physician/allergist/immunologist	3 (2.8%)
Not applicable	51 (47.7%)
Which of the following could have been done to improve your care and experience in the management of CRSwNP? (Allowed to choose more than one answer)	
The availability of more management options for CRSwNP	74 (74.7%)
Smell testing	39 (39.4%)
More time with the clinician	29 (29.3%)
Mental health support	24 (24.2%)
Other	14 (14.1%)

**Table 3 jcm-12-05367-t003:** Qualitative data from participants on satisfaction with different clinicians.

	High Satisfaction Scores	Low Satisfaction Scores
Rhinology Specialist	‘She was thorough in her examinations, concise in explanations and treatment options…I feel they understand my condition.’ (Female, 56–70, USA) ‘Both sinus surgery and follow-up treatment have changed my life for the better…’ (Female, 56–70, East Midlands, UK) ‘A joy to be seen by someone who understood and took a real interest.’ (Female, 56–70, West Midlands)	‘It’s all very reactionary, not proactive, never given any preventative advice, more just live with it until it gets so bad you need an operation again.’ (Female, 56–70, North West England) ‘You have to wait weeks for a CT scan…weeks to go back for the results…months for an operation. In the mean time you can’t sleep, eat or function properly and are in constant pain.’ (Female, 56–70, North West England) ‘Ability to smell isn’t their focus, it’s the ability to breath.’ (Female, 26–40, USA) ‘The last one said “well you won’t really miss it after that long, will you?” No one would say that if you were blind or deaf.’ (Female, 26–40, South East England)
General ENT Clinician	‘My ENT doctor opened my eyes to the world of sinus health and surgery, showed me how much life I was missing.’ (Male, 26–40, USA) ‘Whilst smell and taste has not returned after surgery, overall wellness has improved. Both before and after surgery all ENT specialists have always been sympathetic and professional when dealing with my anosmia.’ (Male, over 70, North East England)	‘They give their best support but the treatments in existence simply aren’t great…’ (Female, 26–40, USA) ‘I have moved abroad, best experience in the Netherlands, Germany has been 50:50. UK I felt was very poor.’ (Male, 26–40, Greater London)
Respiratory physician/ Allergist/ Immunologist	‘My healthcare team has been transparent and I’ve finally felt people are taking my symptoms seriously.’ (Male, 26–40, USA)	‘It is like they don’t really care not realise how exhausting it is.’ (Female, 41–55, Canada) ‘Told to just live with it…nothing he can do.’ (Female, 41–55, South East England)
General Practitioner (GP)	‘GP was sympathetic, nothing he could do and referred me.’ (Male, over 70, North East England) ‘My current GP is the first to take my symptoms seriously. He is an asthmatic and lifetime sufferer of allergies so he related to my condition.’ (Male, 26–40, USA)	‘Not knowledgeable and doesn’t understand living with the condition.’ (Male, 41–55, South West England) ‘For many years the GP has taken a view that since it cannot be cured, no follow up is necessary. All they do is prescribe Flixonase.’ (Male, over 70, Yorkshire and Lincolnshire) ‘GP doesn’t seem to care that I can’t smell. My physical symptoms have improved so my GP hasn’t been helpful trying to help me get my sense of smell back.’ (Female, 26–40, Republic of Ireland)
Other		‘Without a regular doctor, one gets poor, inconsistent support.’ (Female, over 70, Canada)

**Table 4 jcm-12-05367-t004:** Other medical conditions that affect olfaction.

Question	Frequency (%)
Do you have any of the following additional medical conditions that can affect the sense of smell?	
Other sinonasal conditions (e.g., Allergies, nasal tumor, olfactory cleft stenosis)	28 (25.7%)
Not applicable	5 (4.6%)
Smell loss after an operation	6 (5.5%)
Smell loss after a viral infection (e.g., Post COVID-19)	3 (2.8%)
Smell loss after a head injury	1 (1%)
Congenital smell loss (smell loss since birth)	0
Neurodegenerative conditions (Alzheimer’s, Dementia, Parkinson’s Disease)	0
Other	0
None of the above	66 (60.6%)

## Data Availability

All relevant data are contained within the article: The original contributions presented in the study are included in the article, and further inquiries can be directed to the corresponding author.

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
