# Peer review of "Understanding the Impact of Chronic Rhinosinusitis with Nasal Polyposis on Smell and Taste: An International Patient Experience Survey"

_jcm, 2023, doi:10.3390/jcm12165367_

Round 1
Reviewer 1 Report
First, I would like to know the inclusion criteria for the study group, and for a better understanding add a timeline for this article.
Explain if your included patients have other medical conditions, such as allergies, etc.
Line 190-256 please include the report in a table, for a better understanding of the answers.
fig 1 is irrelevant, please replace it with a correlation between the satisfaction and type of medical condition and treatment.
fig 3 - it represents a comparison between the 2 groups? please explain.
table 3 should explain different medical conditions and the results for patients with their pathology. Is there a way to improve? how? please find a way to explain the correlation by statistical evidence of the population analyzed.
minor changes for English.
Author Response
In response to your suggestions and comments. Below are the authors responses and changes.
- Line 101-104 (inclusion and exclusion criteria – ‘Any individual with a self-reported medical diagnosis of chronic rhinosinusitis with nasal polyposis (CRSwNP) were eligible to participate in the survey and were included. Participants who reported that they did not have a diagnosis of CRSwNP were excluded from the study. There were no other exclusion criteria.‘ Line 89-93 (timeline – ‘The concept of the survey was started on 7th November 2022. The final survey design was completed on 30thNovember 2022 after being reviewed by all authors. The survey was distributed through Fifth Sense email newsletters, social media channels and patients presenting to a UK rhinology clinic from 1st December 2022 and survey responses were collated up to and including 1st February 2023.’)
- Line 320-322 (Mentioned in limitations that data on co-morbidities was not collected – ‘There were no strict inclusion or exclusion criteria for respondents to participate in the survey and co-morbidities, allergies or smoking status were not collected.’)
- Please see new Table 3 Line 202-203.
- Figure 1 demonstrates the satisfaction ratings from participants on the management of their CRSwNP and the management of smell and taste impairment related to CRSwNP. It shows reduced satisfaction scores in smell and taste compared to the overall management of CRSwNP. In the survey, participants were not asked to give satisfaction scores on particular treatments for CRSwNP. When participants were asked what treatments, they were currently on and what treatments they found most effective, respondents could choose more than one management option. CRSwNP is the only condition with satisfaction scores (main focus of the study).
- Line 239-243 (explanation for comparison between current treatment and effective treatment groups – ‘Fig 3 compares regular therapies and their perceived limited efficacy in symptomatic relief. Whereas, oral steroids and sinus surgery are effective management options that cannot be used on a regular basis due to side effects and potential serious risks and complications. This may be explained by respondents having severe CRSwNP that is non-amenable to regular medical therapy.’)
- Further subgroup analysis of participants who declared they had additional medical conditions was not undertaken as per point 2. The numbers with additional causes for olfactory dysfunction are small (smell loss after an operation, smell loss after a viral infection, smell loss after a head injury) compared to those with CRSwNP alone and therefore no meaningful comparisons can be drawn. This would be a limitation to the study and this is highlighted in line 320-327 (‘There were no strict inclusion or exclusion criteria for respondents to participate in the survey and co-morbidities, allergies or smoking status were not collected. Conditions that affected smell and taste were collected and grouped together. For example, sinonasal conditions included allergies, olfactory cleft stenosis and sinonasal tumours. However, a subgroup analysis was not performed for these participants as confounding factors were not controlled. A comparison between the responses from the CRSwNP and CRSwNP with additional medical conditions that affected smell groups could have identified differences in perceptions on CRSwNP and olfaction.’). However, it does emphasise that the majority have CRSwNP as the main cause of their olfactory dysfunction.
Reviewer 2 Report
The authors investigate the impact of CRSwNP on smell and taste. Olfactory disturbance affects patients’ quality of life and should be addressed, and this manuscript shows some important points. The following are suggestions for this manuscript.
1. Line 131-132. The majority of the patients were female, although some reports showed more male patients had CRSwNP (https://doi.org/10.1016/j.jaip.2019.05.009). This may affect the result and it would better to be mentioned in the limitation section.
2. Line 145-146. The rates of patients diagnosed by ENT doctors looked different from those in Table 2. This might be because the patients were allowed to choose more than one answer. It would be better to unify the rates in sentence and table. The same thing is seen in Table 3.
3. Line 297-304. The authors compare the patients’ symptoms before diagnosis and after treatment. If there are some data of symptoms before treatment (but after diagnosis) and after treatment, the data will show the efficacy of therapies more intelligibly.
4. Line 381-382. The survey title is ‘Survey of the Impact of Chronic Rhinosinusitis with Nasal Polyposis on Smell and Taste’, and this might affect the results. Even so, hyposmia/anosmia is debilitating symptom of CRSwNP. It would be better to mention the title of the survey.
5. Line 421-422. Is there any in progress clinical trials of biologics for CRSwNP in the UK? If so, it would be better to mention the trial here.
6. Line 434. What is ‘personal safety support’?
Author Response
In response to your comments and suggestions, below are the individual point responses by the authors:
- Line 318-320 ‘Moreover, the majority of respondents were female despite there being evidence that more male patients had CRSwNP which could affect results (5).’
- Line 147-151 ‘The majority of participants (81.3%) had suffered with symptoms of CRSwNP for over five years with 69.2% being formally diagnosed for over five years. 97.2% were diagnosed by a general otorhinolaryngology (43.1%) or otolaryngology clinician specializing in rhinology (36.9%); 66 participants (61.7%) declared they were not under regular follow-up.’ – addressed Table 2
Line 249-253 ‘The majority (60.6%) of total respondents only had CRSwNP with 5 respondents providing no answers. 38 participants (35%) declared they had other conditions that affected their sense of smell. Of these, 73.7% had other sinonasal conditions that included allergic rhinitis, nasal tumours or olfactory cleft stenosis, 15.8% after an operation and 7.9% after COVID-19/viral infection.‘– addressed Table 4 (previously table 3 on last version)
- We think patient symptoms before diagnosis is equivalent to patient symptoms being diagnosed but before treatment as conventionally patients with CRSwNP would be initiated on treatment once a diagnosis is established. Some may have already been started on treatment prior to diagnosis if they were seen in the community by a general practitioner (eg. Nasal douching and intranasal steroids). Therefore, symptoms before diagnosis demonstrates what patient perceptions are of their symptoms before regular CRSwNP treatment and this is compared to perceptions of symptoms after treatment. Line 258-260 ‘This is to demonstrate the change in perceptions before starting treatment once a diagnosis of CRSwNP has been established and perceptions after regular treatment.’.
- Can you please clarify where you are suggesting we should mention the title of the survey? The title of the survey has been designed to enable respondents to easily understand the main aim of the survey. Hyposmia/anosmia are key symptoms of CRSwNP and the aim of this survey was to demonstrate how much of an impact these symptoms had on quality of life when compared to other key symptoms of CRSwNP (nasal obstruction, rhinorrhoea and facial pain).
- Have mentioned and cited an ongoing UK randomized, double-blind, parallel group, clinical trial – ANCHOR-1 for CRSwNP. Line 393-394 ‘…clinical trials such as the randomized, double-blind, parallel group, phase III study, ANCHOR-1 (16,33,34)’(reference 34)
- Line 406-407 ‘…personal safety support such as, improving awareness on gas safety and rotten food through education and support groups…’
Round 2
Reviewer 1 Report
the manuscript is improved, well done.
minor spelling errors.
Author Response
Thank you. Spelling errors have been corrected in the manuscript.
Reviewer 2 Report
4. There might be a selection bias where patients with smell/taste disturbance tend to participate in the survey. The bias affects the result of Table 2. It would be better to mention the bias in the limitation section.
Author Response
4. I have added this in the limitations section. Line 295-297 'Respondents with strong opinions or experiences with CRSwNP and those particularly affected by smell or taste disturbance would be more likely to participate.'